# Conserved Patterns in Developmental Processes and Phases, Rather than Genes, Unite the Highly Divergent Bilateria

**DOI:** 10.3390/life10090182

**Published:** 2020-09-06

**Authors:** Luca Ferretti, Andrea Krämer-Eis, Philipp H. Schiffer

**Affiliations:** 1The Pirbright Institute, Ash Road, Pirbright, Surrey GU24 0NF, UK; 2Big Data Institute, Nuffield Department of Medicine, University of Oxford, Oxford OX3 7BN, UK; 3Institut für Genetik, Universität zu Köln, Zülpicher Straße 47a, 50674 Köln, Germany; a.kraemer.eis@googlemail.com; 4Institut für Zoologie, Universität zu Köln, Zülpicher Straße 47b, 50674 Köln, Germany

**Keywords:** evolution, bilateria, gene ontology, expression profile, development, evodevo, orthology

## Abstract

Bilateria are the predominant clade of animals on Earth. Despite having evolved a wide variety of body plans and developmental modes, they are characterized by common morphological traits. By default, researchers have tried to link clade-specific genes to these traits, thus distinguishing bilaterians from non-bilaterians, by their gene content. Here we argue that it is rather biological processes that unite Bilateria and set them apart from their non-bilaterian sisters, with a less complex body morphology. To test this hypothesis, we compared proteomes of bilaterian and non-bilaterian species in an elaborate computational pipeline, aiming to search for a set of bilaterian-specific genes. Despite the limited confidence in their bilaterian specificity, we nevertheless detected Bilateria-specific functional and developmental patterns in the sub-set of genes conserved in distantly related Bilateria. Using a novel multi-species GO-enrichment method, we determined the functional repertoire of genes that are widely conserved among Bilateria. Analyzing expression profiles in three very distantly related model species—*D. melanogaster*, *D. rerio* and *C. elegans*—we find characteristic peaks at comparable stages of development and a delayed onset of expression in embryos. In particular, the expression of the conserved genes appears to peak at the phylotypic stage of different bilaterian phyla. In summary, our study illustrate how development connects distantly related Bilateria after millions of years of divergence, pointing to processes potentially separating them from non-bilaterians. We argue that evolutionary biologists should return from a purely gene-centric view of evolution and place more focus on analyzing and defining conserved developmental processes and periods.

## 1. Introduction

Bilateria comprise about 99% of the extant eumetazoans [1] and are classified into 32 phyla [2]. The taxon “Bilateria” has been defined based on morphological key innovations, namely bilateral symmetry, triploblasty, an enhanced nervous system and a complex set of cell types [3]. Probably the most striking observation is the existence of a wide variety of body plans, accompanied by high morphological diversity in many bilaterian phyla [4]. It is thought that the major bilaterian phyla emerged from their stem group in a fast radiation during the early Cambrian, about 540 million years ago [1], but the appearance of the common bilaterian ancestor, and its exact timing, is controversial [5,6,7,8]. It is also difficult to define unifying morphological properties for larval or adult Bilateria, as their descendants underwent extensive re-modellings (including secondary reductions and simplifications) of body forms in the lineages leading to extant crown clades. Nevertheless, the general process of development from single cell to adult is conserved among all animals. Recently, support for the hypothesis that the developmental transcriptome might be a conserved trait across diverse groups of animals has been found [9,10]. In particular, several studies described a conserved phylotypic period mid development based on transcriptomic analyses [11,12,13], akin to the morphological hourglass model of developmental progression [14,15], itself an extension of von Baer’s reverse funnel model of development in animals [16].

Despite these similarities in the global transcriptome, comparisons at the molecular level reveal that many developmentally important genes are older than the ancestor of Bilateria. Among them are determinants of dorso-ventral and anterior-posterior patterning, eye formation, segmentation, and heart development [5,17,18,19,20]. In a recent study, it was estimated that up to 85% of the genes present in any Bilateria examined were already present in the ur-bilaterian [21], and only about 15% are of more recent origin. Deep comparative genomics including non-bilaterian metazoans, also revealed that the major developmental signaling pathways are already present in cnidarians, ctenophores, placozoans and poriferans [22,23,24], and must predate the origin of Bilateria. At the same time, some analyses suggested the existence of genes specific to large clades with divergent sub-taxa, for example the deuterostomes [25]. These genes would then define the corresponding clade, for example setting Bilateria apart from their non-bilaterian ancestors, thus making them somehow biologically “special”. However, the importance of specific conserved genes is often overestimated. Genes can be lost, duplicated and change their function without prejudice to the original biological function, as long as there is sufficient functional redundancy in the corresponding regulatory networks and pathways. For this reason, we decided to test the hypothesis that specific genes define large biological taxa, as the Bilateria, considering also the opposite one: biological function and processes (e.g., the conserved developmental transcriptome), rather than specific genes unite taxa.

Using reciprocal blast searches, orthology pipelines, and an array of filtering steps we first identified a set of orthologues proteins that satisfied our initial criteria for Bilateria specificity. We then secondly tried to disprove the bilaterian specificity of these proteins with high-confidence secondary validation. Re-analyzing an initial set of 85 orthogroups with new data from the ever growing database of available proteins we found only 35 orthogroups to potentially remain “Bilateria-specific”. It must be assumed that this set would shrink even further with more data becoming available, in particular from more non-bilaterians.

Despite the limited confidence in bilaterian specificity of our gene set, we observed an interesting signal in the data. Analyzing the set of 85 orthogroups in more detail we found that these genes are not only retained in species separated by a billion years of independent lineage evolution, but that most of them act in key developmental processes. These genes show very similar and peculiar expression profiles across analogous developmental stages in highly divergent model species in the fishes, the arthropods, and the nematodes. This finding supports the idea that evolutionary changes in the genetic machinery of development were coupled to the emergence of Bilateria and are still conserved in very divergent taxa today.

## 2. Materials and Methods

Using a comparative genomic approach, we aimed to identify genes shared among and specific to Bilateria. To represent the spectrum of the different bilaterian clades we selected ten species from the three major clades-Deuterostomia, Ecdysozoa and Lophotrochozoa-concentrating on species with a fully sequenced and annotated genome (Table 1). We included more species from Lophotrochozoa than from the other two clades, since well-annotated complete genome sequences from lophotrochozoan model organisms were still under-represented in public databases at the beginning of our studies. Our selection is more unequal within clades, with an over-representation e.g., of vertebrates, insects, molluscs and annelids, reflecting the availability of well-annotated genomes at the time of sampling. Our data include three model species (*D. rerio*, *D. melanogaster*, *C. elegans*) on which our functional and expression analyses are based. To obtain a high contrast between Bilateria and non-Bilateria, we used seven non-bilaterian species, as in a previous study [26]. Proteomes and data from InterProScan [27] and the Gene Ontology consortium [28] were retrieved from the sources cited in Table 1.

### 2.1. Initial BLAST and Clustering

To find orthologs we set up a stringent analysis pipeline. We first species blasted all-versus-all among the 17 proteomes using the stand-alone BLAST version 2.2.25+ (NCBI, Bethesda, MD, USA) with a cutoff of $E=10^{-5}$. We then discarded all bilaterian proteins which had a significant hit in any of the non-bilaterian sequences. Please note that significant hits with non-bilaterians could be due to shared domains in otherwise non-homologous proteins, therefore resulting in the exclusion of some truly bilaterian-specific genes. This filtering resulted in 13,582 candidate proteins. From these we constructed orthologous groups using two different ortholog finders, InParanoid and OrthoMCL [29,30], using default parameter settings. Both programs were rated highly in benchmarking studies which analyzed the performance of orthology-prediction methods [31,32]. For the purpose of detecting orthologs of very diverged species, we chose OrthoMCL version 2.0.2 [29] and the associated MCL version 11-335, as they were shown to be more robust [31,32] than similar programs. Finally, we grouped the ortholog clusters into four sets (*A*, *L*, *M*, *C*; Figure 1), satisfying different conditions on how widely orthologues are shared across Bilateria. Due to the poor sequence quality, we treated the Lophotrochozoon *A. californica* differently: where available we included *A. californica* orthologues, but we did not require them to be present in set *A* (“all species”).

The above data sets are ordered by degree of initial confidence in the bilaterian specificity of the respective proteins. At the same time, they reflect the level of conservation of the proteins across different species. Seeking to be conservative regarding bilaterian specificity on the one hand (set *A* being most conservative), but on the other hand to analyze a dataset which is as comprehensive as possible, we focused our further analyses on the intersection of sets *L* and *M* (called $L^{\prime }$; Figure 1) comprising 85 clusters.

### 2.2. Validation with BLAST, OMA, and Phylogeny

As our original dataset was bilaterian and non-bilaterian species was assembled prior to 2012 and the steady flow of newly sequenced species has added massive new numbers of protein predictions to online databases we wanted to re-evaluate the potential bilaterian specificity of our 85 final clusters. To this end we downloaded 19,185,382 bilaterian and 497,273 non-bilaterian proteins from the NCBI database in August 2017 and created individual databases for taxa in bilateria and non-bilateria. Next we used Diamond [33] with the “–very-sensitive” parameter to blast the *D. rerio*, *C. elegans*, and *D. melanogaster* proteins for each of the 85 clusters against the individual taxa. Additionally, we used NCBI-BLAST with the parameters described to be most sensitive in finding orthologs in [34] for the non-bilaterian taxa. For each of the 85 clusters we then collected the 10 best blastp hits per taxon. Along with the original model organism proteins we obtained 96,287 proteins in this way, which we used as input for the OMA orthology-prediction pipeline [35]. From the OMA output we selected hierachical cluster of orthologues (HOG) for those of the original 85 clusters where the HOG contained non-bilaterian proteins. Finally, we aligned all sequences per HOG with clustal-omega [36] and reconstructed a phylogeny using IQTree with automated model selection and 1000 fast bootstrap replicates [37]. We then analyzed these trees by eye.

### 2.3. Selecting Most Conserved Orthologs

For the three model species *D. rerio*, *D. melanogaster*, and *C. elegans* we extracted the UCSC tracks of basewise PhastCons conservation scores [38] calculated across insects, teleosts and nematodes, respectively. We used these scores to rank all genes in a cluster according to their fraction of strongly conserved sites, i.e., sites with PhastCons score >0.99. We selected the highest-ranking one from each cluster as the “Most Conserved Ortholog” (MCO). These genes likely retained the same function, even in highly diverged species – a conjecture based on the idea that strong conservation reflects long-term evolutionary (and functional) constraint and that neo- or sub-functionalized paralogs tend to be less conserved. We used the fraction of strongly conserved sites, instead of the average conservation score, since we are interested in the degree of conservation in function, not in sequence. We reasoned that functional conservation should be related to high conservation of alleles at functional sites, while the remaining bases can evolve fast. However, the two ways of measuring similarity are highly correlated on a genome-wide scale across all coding exons (r=0.91 for *D. melanogaster*, 0.84 for *D. rerio* and 0.71 for *C. elegans* (Figure 2)). We performed all subsequent analyses on both sets: the set of “All orthologs and paralogs” (AOs) and the set of MCOs.

### 2.4. Multi-Species Gene Enrichment Analysis

To examine function of the genes in set $L^{\prime }$ by *in silico* methods, we obtained their gene ontology (GO) terms for the three model species from the GO database [28]. To identify terms enriched in COP genes compared to the genomic background, we employed two strategies: the conventional single-species enrichment analysis based on Fisher’s exact test and a novel method, termed multi-species gene enrichment analysis (MSGEA). Standard single-species Fisher’s exact tests for enrichment cannot capture a signal of moderate, but joint, enrichment across different species. Instead, with MSGEA we compute an exact *p*-value for a pooled set of species, employing the following steps. First, for each GO term we counted the number of descendants in the GO tree. Then, we computed the *p*-values for the terms occurring in any species and corrected for multiple testing [39]. More specifically, let $n_{go,s}$ be the count (i.e., number of occurrences) of the GO term go in species *s* contained in our set and let $N_{go,s}$ be the count for the whole genome. Furthermore, let $n_{s}=\sum_{go}n_{go,s}$ and $N_{s}=\sum_{go}N_{go,s}$. Assume that all species under consideration diverged at the same time and evolved independently after that (i.e., they have a starlike phylogeny). The null hypothesis of the MSGEA test is that a given GO term was not over-represented in the common ancestor. Therefore, a significant *p*-value means that the genome of the ancestor was already enriched in the GO term considered. Being conservative, assume that the null distributions of $n_{go,s}$ are independent hypergeometrics with parameters $N_{go,s}$, $n_{s}$ and $N_{s}$ in each species. The enrichment statistics $X_{go}$ is then the sum of the normalized enrichments of $n_{go,s}$ across species, i.e., the sum of the *z*-scores:$$X_{go}=\sum_{s}\frac{n_{go,s}-E\left(n_{go,s}\right)}{\mathrm{SD}(n_{go,s})}∼\sum_{s}\frac{n_{go,s}-N_{go,s}n_{s}/N_{s}}{\sqrt{N_{go,s}n_{s}/N_{s}}}$$
where a Poisson approximation is used to define the score. Finally, the *p*-value is the probability
$$p_{go}=\mathrm{Prob}\left(x\geq X_{go}\;|\;N_{go,s},n_{s},N_{s}\right)\;,$$
where the distribution of *x* follows from the hypergeometric distributions of the $n_{go,s}$ with the above parameters.

For a single species, this test coincides with the standard one-tailed Fisher’s exact test for GO enrichment, and therefore is consistent. The exact estimation of *p*-values is computationally intensive; an optimized code, written in C, is available from the authors upon request. Since MSGEA is based on the hypothesis of independent evolution of each lineage, it can be applied only to starlike phylogenies. For the situation considered here, this assumption is met, since the “roundworm-fruit fly-zebra fish” phylogeny is approximately starlike. We find that a considerable fraction (30–70%) of significant GO terms in our data set is detected by MSGEA, but not by single-species enrichment.

### 2.5. Detailed Functional Analysis of Genes from Model Organisms

To mine literature databases for functional studies in model organisms with a focus on development we employed biomaRt: we used the Bioconductor module [40] biomaRt 2.19.3 (http://biomart.org) to extract information for proteins present in cluster set $L^{\prime }$. Wormbase release WS220 was queried for *C. elegans* proteins and ENSEMBL 75 for *D. melanogaster* and *D. rerio* proteins. We then searched the literature for experimental validations of protein function. We grouped the results in six major categories, described in Results and Discussion and summarized in Appendix A. These categories represent prominent molecular functions during embryogenesis and development. Based on the retrieved annotations we assigned proteins in set $L^{\prime }$ to these categories.

### 2.6. Cross Species Expression Profiles

We retrieved expression data for different developmental stages in *D. melanogaster* [41,42], *D. rerio* [43] and *C. elegans* [44]. *D. rerio* data for adult stages were not used. The expression profiles were transformed by taking the logarithm to base 10 of the expression levels and then subtracting the $\log_{10}$ of the mean expression at each stage.

We devised the following test to find characteristic expression patterns: first, we computed Pearson’s correlation coefficient between expression profiles for different genes. For each profile in each set of the genes from 85 COPs, we performed two types of tests: (i) a Mann-Whitney test on the distribution of correlations, comparing the correlation of the profile with other bilaterian genes versus the correlation with genome-wide profiles, in order to detect profiles that are more correlated with the ones of other bilaterian genes than with the rest of the genome; (ii) for each profile, we classified the remaining genes as “highly correlated” or “not highly correlated” in expression, using correlation thresholds of r=0.5, 0.7 and 0.9; then, we tested for enrichment of correlated profiles among bilaterian genes by Fisher’s exact test.

A Benjamini–Hochberg correction to control for false discovery rate in multiple testing procedures was applied to the *p*-values resulting from these tests. All the significant profiles (p<0.05) were clustered based on their correlation coefficients *r* by complete-linkage hierarchical clustering implemented in the R statistics software package, using 1−r as distance measure and selecting clusters at height h=0.75. After clustering, profiles were normalized by their average expression across stages, then averaged across each cluster. The resulting profile shapes are shown in row 2 of Figure 3.

### 2.7. Are Expression Profiles Driven by Gene Function?

We performed a randomization test to explore if the mean (logarithmic) expression level of COPs genes could be explained by their function. For each developmental stage we checked if expression correlates with that of randomly selected genes possessing equal or similar GO terms, i.e., we checked if genes of different age, but similar ontology, would show similar expression profiles. We performed this analysis only on set $L^{\prime }$.

For this purpose, we randomly sampled from the whole genome 100 sets with the same number of genes as are contained in set $L^{\prime }$. We did this in two different ways. (i) The first (“random background”) was a random sampling of genes from the genome. After sampling, we computed the mean normalized expression of the selected genes for each stage. We repeated this procedure 100 times, and thus obtained a distribution of normalized means, represented as boxplots (in white) in Figure 3, Appendix A. (ii) The second was a “matched-function” sampling: first, for each GO term in the original set, we listed all genes with the same annotation and extracted from this list several random genes equal to the number of occurrences of the term; second, we pooled all the resulting lists; third, we extracted from this list several random genes equal to the number of genes in the original set. This way, we obtained sets of the same size and function (approximately, at least) as our original set. We applied this to the whole set $L^{\prime }$ and for all subsets corresponding to the six functional categories described above (Appendix A).

### 2.8. Age Index of Proteins

We used the phylostratigraphy from Domazet-Lošo et al. [43] for *D. melanogaster*, *D. rerio* and *C. elegans* to define two age-groups of proteins: (i) older proteins, with an origin in phylostrata 1 to 6; (ii) younger proteins, with an origin in phylostrata 8 and higher. Proteins that we identified as bilaterian-specific (i.e., the ones in our sets *A*, *L*, *M* and *C*) were excluded from the two groups, irrespective of their original phylostratigraphic classification. We repeated all the analyses described in the previous sections, and compared separately our COPs genes with the genome-wide background of older and of younger genes (Figure 3).

## 3. Results

### 3.1. Defining Potential Clade-Specific Proteins in Bilateria

To identify genes that potentially newly emerged in the “ur-bilaterian” (Figure 1a) and have been retained in its descendant species our initial approach was grounded in a comparison of ten bilaterian and seven non-bilaterian species with fully sequenced and annotated genomes as of 2011. We downloaded 383,586 protein sequences in total. About 70% (268,252) are from Bilateria, covering the protostome super-groups Lophotrochozoa and Ecdysozoa, and the Deuterostomia, and about 30% (115,334) are non-bilaterian (Table 1).

We discarded all bilaterian sequences that had a best BLAST hit below the threshold *E*-value of $10^{-5}$ with non-bilaterian sequences. We retained 13,582 bilaterian-specific candidate proteins which were grouped into 1867 clusters of orthologous proteins (COPs) using the OrthoMCL pipeline [29]. We condensed the raw set of 1867 clusters of orthologous proteins by applying filters with different stringency for the taxa included (Figure 1) and obtained the following four sets:*C*:Each cluster must contain one or more representatives from each of the three major **c**lades, Lophotrochozoa, Ecdysozoa and Deuterostomia. This set contains 506 clusters.*M*:Each cluster must contain representatives from all **m**odel organisms, *D. rerio*, *D. melanogaster* and *C. elegans*. This set contains 160 clusters.*L*:Each cluster contains representatives from all major clades, as in set *C*. However, only those clusters are permitted for which the representation of species is explained by at most one **l**oss event along the species tree (set *C* does not have this restriction). This resulted in 125 clusters*A*:Each cluster contains representatives of **a**ll bilaterian species considered (no protein loss is admitted) except possibly *A. californica*. This set has 34 clusters.

On average, each of the nine bilaterian species is represented in 66.2% of the 506 ortholog clusters in set *C*. In set *M*, this percentage is 80.9%, and in set *L* it is 91.2% (see Table 2), reflecting the levels of stringency of the filtering criteria.

### 3.2. Orthologues Conserved Across Divergent Bilateria

In our analysis, we included the highly divergent bilaterian model organisms *Caenorhabditis elegans*, *Drosophila melanogaster*, and *Danio rerio*. These have very well curated and annotated genomes and are therefore helpful to find genes which are also functionally conserved over 500Myrs of independent evolution. To take advantage of this, we decided to use the intersection of sets *M* and *L* (termed $L^{\prime }=L\cap M$) for further analysis. Compared to set *L* (125 clusters), this intersection lacks 31 clusters missing a *C. elegans* ortholog and 9 clusters missing an ortholog from *D. melanogaster*, resulting in a set of 85 orthogroups. Except for *A. californica*, the least represented species in these clusters is *S. purpuratus* (genes of this species occur in 63 of 85 clusters). By construction the model organisms (*C. elegans*, *D. melanogaster*, *D. rerio*) are represented in all clusters in $L^{\prime }$, thus unifying highly diverse bilaterian species. On average, a (bilaterian) species is represented in 93.1% of the clusters.

### 3.3. Clade-Specificity Declines with Data Availability

A general drawback of our experimental procedure is its reliance on correctly identified and annotated genes. Since most of the organisms in our study are non-model organisms, they may suffer from incomplete or erroneous gene annotation. For example, we omitted the bilaterian species *A. californica* from downstream analyses because of its low-quality protein annotation. It is also possible that truly existing genes are missing from the non-bilaterian dataset due to annotation or prediction errors or simple insufficient availability of sequenced species, leading to wrong inferences of bilaterian specificity. To test this possibility, we first verified by additional BLAST searches at NCBI that none of the 85 COPs in set $L^{\prime }$ had BLAST hits below an *E*-value of $10^{-5}$ in non-bilaterians or other eukaryotes available in August 2014. To accommodate to the very fast growing amount of data from across the bilateria and non-bilateria becoming available in recent years, we decided to implement a second, more comprehensive test to validate the potential bilaterian specificity set $L^{\prime }$. Conducting searches with Diamond and NCBI-BLAST against a wide array of protein data downloaded in August 2017 from NCBI we identified potential homologs for proteins in the 85 COPs. Using OMA and phylogenetic inferences we could then identify non-bilaterian orthologs for all but 35 of the 85 set $L^{\prime }$ COPs. This showed that with additional genomes and transcriptomes becoming available, the number of potentially bilaterian-specific orthologous groups identified in this way could decrease drastically.

#### Most Conserved Orthologues

Although not being strictly bilaterian-specific, we regarded the set $L^{\prime }$ COPs as biologically interesting since they are conserved across huge evolutionary distances with many speciation events separating them in the three model organisms (and additionally divergent enough from non-bilaterian sequences to not be detected in our first re-BLAST screen). We thus wondered what their biological function is and if this function could be correlated with a distinct process in the biology of Bilateria. As a result of our experimental design, but also due to different duplication histories, all COPs contain differing numbers of paralogs from different species. On average, we found most paralogs in Deuterostomia, followed by Ecdysozoa and Lophotrochozoa. The highest average paralog number is found for *D. rerio*, which might be a consequence of the additional genome duplication events in Teleosts [45]. Since the function of paralogs might differ from the function in the last common ancestor, we constructed an additional set of ortholog clusters free of paralogs: we extracted from each cluster in our original set $L^{\prime }$ the most conserved ortholog for each species, according to the PhastCons conservation score [38], and discarded the less conserved paralogs. This reduction does not affect the number of clusters, but the number of proteins within a cluster. In our downstream analyses we considered both versions, the “most conserved orthologs” (MCO) and the “all orthologs” (AO) catalog for each cluster.

### 3.4. Gene Ontology and Functional Classification

#### 3.4.1. Go Terms Reveal a Link to Development

To gain an overview over the biological functions of the proteins in set $L^{\prime }$, we extracted and analyzed their associated GO terms (summarised in Appendix A). To quantify simultaneous enrichment in multiple species, we developed a novel generalized Fisher’s exact test for multiple species, which we term MSGEA (“Multi Species GO-Enrichment Analysis”; Figure 4 and Appendix A). In contrast to standard GO-enrichment analysis, MSGEA does not solely focus on the over-representation of GO terms in single species, but is able to detect GO terms enriched across several species, even if not over-represented in any single species. Of special interest for our analysis were GO terms occurring across all three model organisms: such terms may indicate the conservation of biological function across large evolutionary time scales. To our knowledge, this novel method is the only test that is sensitive to coherent enrichment of GO terms across multiple species (see Methods).

Applying MSGEA we extracted a list of GO terms which are likely associated with long-term retained functions of our candidate genes. Concentrating on the domain “biological process” of the GO database, we find the terms “development”, “muscle”, “neuron”, “signaling” and “regulation” to be strongly over-represented (Figure 5 and Appendix A). The terms identified through in silico analysis suggest a prominent role for these orthologues in development of the bilaterian nervous system, as well as musculature, morphology, and cell-cell signaling.

#### 3.4.2. Inference of Biological Function Through Literature Mining

Since orthology is a strictly phylogenetic criterion, and since GO classification draws heavily on sequence homology, without necessarily reflecting conserved function, we decided to perform an in-depth human-curated literature search to collect additional evidence for the functional role of genes in COP clusters. We mined the literature databases and compiled human-curated functional descriptions, based on experimental evidence, for the proteins contained in set $L^{\prime }$. We defined the six classes “Neuron related”, “Morphology related”, “Muscle related”, “Signaling related”, “Regulation related” and “Others” (Appendix A). We used functional descriptions extracted from the literature to assign each of the 85 set $L^{\prime }$ clusters to one of these six classes (Figure 5). The manual extraction of species- and protein-specific functional information allowed us to compare and better interpret functions across the three model organisms.

To compare this human-curated classification with the one based on the GO database, we assigned all GO terms appearing in set $L^{\prime }$ to one of the six classes above. Since a given protein may be associated with many GO terms there is no one-to-one relationship of the human-curated annotation and the GO terms. For example, while the human curation assigns eight of the 85 clusters (the left-most columns in Figure 5) to the category ‘Morphology related’, morphology related GO terms occur in 33 clusters, scattered throughout the six classes. Similarly, signaling related GO terms are found in almost every cluster, while the class “Signaling related” contains only 18 clusters. However, our manual curation is still broadly consistent with GO annotations. For example, GO terms associated with “muscle” do occur in the class “Muscle related” and GO terms related to “neuron” are mostly found in the “Neuron related” clusters.

### 3.5. Expression Profiles

To analyze expression of genes in set $L^{\prime }$ across developmental stages and to compare it between model organisms, we used publicly available data for *D. melanogaster*, *D. rerio* and *C. elegans*. We normalized expression values-separately for each species and for each developmental stage-by the genome-wide average, and then log-transformed them.

#### 3.5.1. Pooled Profiles

To obtain a global picture of the expression profiles of genes in set $L^{\prime }$, we calculated mean and median profiles for each of the three model organisms, shown as orange lines in Figure 3 and Appendix A, top row. To see how expression of widely shared bilaterian genes compares to background expression, we generated two different background sets for each species. The first is a random background, obtained by randomly selecting genes from the model organism genomes and recording their expression (distribution shown as white boxplots in Figure 3). The second background, which we call the ‘matched-function’ background (orange boxplots), is a random collection of genes with GO terms matched to those found in set $L^{\prime }$. The background distributions were constructed independently for each stage and for each set, and they may differ in size and content. We grouped the developmental stages from fertilized egg to the adult body into coarse grained, yet comparable, phases: “Blastoderm”, “Gastrulation and Organogenesis”, “Hatching to Larva” and “Adult” (separated by vertical lines in Figure 3).

For all three species, the set $L^{\prime }$ profiles roughly follow the shape of the background profiles across embryogenesis, except at a higher expression level (Figure 3). In *D. melanogaster*, expression of widely shared bilaterian proteins is initially low, then rises until “dorsal closure”, after which it slightly decreases again. In *C. elegans*, expression is also initially low and then rises towards “ventral enclosure”. This pattern is mirrored in *D. rerio*, where expression peaks around the pharyngula stage. The profiles seen in *C. elegans* and *D. rerio* are congruent with the major cycle of tissue proliferation and differentiation during organogenesis in these species [46]. During ‘dorsal closure’ in *Drosophila*, “ventral closure” in *C. elegans* and “pharyngula” in *Danio*, the head, the nervous system and the bilaterally organized body develop [47]. In *Drosophila* we find a second expression peak during a second cycle of proliferation corresponding to metamorphosis. The matched-function profile largely resembles the profile of set $L^{\prime }$. However, set $L^{\prime }$ genes show distinctly low expression in the very early embryonic stages, a pattern which is consistently seen in all species (Figure 3 and Appendix A). Another common feature is that expression of MCO genes is consistently higher than of AO genes (Figure 3).

#### 3.5.2. Characteristic Expression Profiles

To further characterize expression profiles specific for genes widely shared among Bilateria, we aimed at identifying characteristic profiles in our gene set. Methods exist to identify tissue-specific expression profiles, or to differentiate between profiles, but no standard techniques are available to detect enrichment of profiles across a set of genes. To overcome this limitation, we employed an ad hoc “profile enrichment” strategy for each species separately. Briefly, we considered a COP expression profile to be characteristic of Bilateria, if we found that similar expression profiles were significantly more abundant among genes in our COPs than among the rest of the genome (for details, see Methods). We extracted all such statistically significant profiles, then clustered individual profiles based on their similarity and distilled representative ones for each species (Figure 3, second row).

For each of the three model species qualitatively similar patterns are retrieved: an intersection of increasing expression profiles (yellow and green lines in Figure 3, 2nd row) with slightly decreasing profiles (dark red and dark blue lines). The intersection occurs right before the developmentally important events “dorsal closure”, “ventral enclosure” and “pharyngula”. A second crossing of profiles can be seen in the fly, just before metamorphosis into the adult body. To interpret this observation, we considered the following functional classification.

#### 3.5.3. Expression Profiles Stratified by Function and Age

We determined the mean expression profiles for the six functional classes-morphology- (8 set $L^{\prime }$ COPs), muscle- (11), neuron- (23), signaling- (18), regulation-related (20) and other genes (5) (Appendix A). We find upward and downward peaked expression patterns for the class’s morphogenesis, neuron and muscle development, which are even emphasized compared to the pooled profiles of the global analysis. This is in particular true for expression before the onset of gastrulation. On the contrary, the genes related to regulation and signaling display a rather constant profile, without the characteristic expression peaks of morphogenesis.

For species as different as animals, plants and fungi, it has been repeatedly observed that timing and level of expression during development depend on the evolutionary age of a protein [43,47,48,49,50]. For instance, one recurrent finding was that older genes tend to show higher expression during early embryogenesis. To examine our proteins in this respect, we analyzed whether the set $L^{\prime }$ expression profiles are more similar to the profiles of younger or of older genes, and we calculated background distributions considering the age of genes. Genes which arose before Bilateria (phylostratigraphic level less than 7, corresponding to Bilateria, according to the strata-numbers by Domazet-Lošo) were defined as “older” and all others (level >7) as “younger” [43]. Expression profiles of genes (AO) in set $L^{\prime }$ match very well the profiles of the ’younger’ genes with matched function (red lines and orange boxplots in Figure 3 and Appendix A). All three species show low expression of young and of set $L^{\prime }$ genes during early development. Hence, this late onset of expression appears to be an evolutionary novelty of Bilateria. This is confirmed by a direct comparison with profiles of genes at phylostratigraphic level 6, corresponding to Metazoa (Appendix A). After “dorsal closure” in *D. melanogaster*, “ventral closure” in *C. elegans*, and “pharyngula” in *D. rerio*, expression increases, and older and younger profiles converge.

## 4. Discussion

The most significant innovations of bilaterian animals are a third germ layer (the mesoderm), a liquid filled body cavity surrounded by mesoderm (the coelom), a complex nervous system, and a large number of specialized cell types. The formation and presence of these features had a massive effect on the evolutionary possibilities and success of bilaterians. By analyzing the genomes of bilaterian and non-bilaterian species we aimed at testing whether there is a gene correlate with these morphological key innovations. Roughly a billion years of divergence between the most distant species, chromosome fusions, fissions and re-arrangements, genome duplications, expansions and losses of gene families, as well as myriads of nucleotide substitutions might have blurred the common genetic heritage of Bilateria. However, those genes and proteins which have evolved under steady purifying selection, i.e., play a key non-redundant role in extant Bilateria species, should still be identifiable as orthologs across Bilateria. We tested if a set of genes that potentially evolved in the bilaterian ancestor and is not present in non-bilaterians could be found. We were initially able to detect a set of 85 orthogroups (set $L^{\prime }$), containing genes which are shared between widely divergent Bilateria and, at the same time, appeared to be without orthologues in non-Bilateria. We reasoned that this set had to be taken with caution, as for example there is no clear-cut threshold of sequence similarity which separates orthology from non-orthology. In such cases the decision, whether a gene with partial homology should be included into an orthogroup or not, is somewhat arbitrary. For the 85 set $L^{\prime }$ COPs we have considered partial homology to a non-bilaterian sequence as an insufficient criterion to exclude an orthogroup from our initial list. A similar BLAST strategy as implemented by us, except without subsequent orthology clustering, had been used before to determine the evolutionary origin of genes and to assign phylostratigraphic ages [43,50,51,52]. One pitfall of this approach is that an overall alignment score, such as a Blast *E*-value, may lead to both false positive and false negative predictions. False positives arise from the (wrong) inclusion of a protein into the bilaterian-specific list, which *does* have non-bilaterian orthologs, but which are missed due to low sequence similarity or due to absence from the database. False negative predictions arise from missing truly bilaterian-specific proteins, for instance when a conserved domain is identified in a protein for which a homologous domain is also present in non-bilaterians. One such example is the C2H2 zinc finger protein CTCF which has been described as bilaterian-specific before [26]. Our search did not recover this protein, because multi-zinc finger proteins exist in cnidarians and their conserved Cys/His residues and linker regions evoke a similarity above threshold. As a result, our approach will wrongly treat such proteins as older and not bilaterian-specific and, consequently, eliminate them. Similar errors may affect other proteins with extended conserved domains. It thus remains possible that a different (potentially partly overlapping) set of bilaterian-specific COPs exists, which our search strategy failed to identify. In contrast, genes may be wrongly assigned to subclades which are younger than bilateria. For instance, this could be a consequence of non-orthologous gene displacement (NOGD) [53], a process in which functions are taken over by new or different proteins and the original orthologue is lost. Finally, we could have wrongly classified a protein as bilaterian-specific in our initial screen because faint homology to more anciently diverged species is missed.

Another limitation in our initial screen was the restriction to 16 high-quality genomes, from which we inferred bilaterian specificity. First, it has now been shown that OrthoMLC, which we mainly relied on, is limited in finding distantly related orthologues if taxon sampling is coarse [54]. This problem could have been inflated by our restriction of candidates in our blast searches before running orthology-prediction pipelines. Secondly, new entries to the databases may change the inferred phylostratigraphic age of an orthogroup. Being aware of this and in particular because sequencing efforts are increasingly directed towards non-standard model organisms in recent years, we have scrutinized and re-inspected our initial candidate set of set $L^{\prime }$ COPs with refined tools and new data. While the remaining sub-set of 35 orthogroups might consist of bilaterian-specific proteins, it is likely that more data from so far not-yet-sequenced non-bilaterian species will also reveal orthologous for some of these proteins. These caveats raise doubts on the likelihood to detect high-confidence genes that are unequivocally bilaterian and at the same time widely retained in this taxon. Our results clearly raise the possibility that there are potentially just a handful of Bilateria-specific genes, or even none. At the same time, our approach of partially falsifying the results of the first orthology screen by implementing two additional checks shows that if there is any such gene, careful and elaborate measures have to be taken to validate potential candidates. At the very least all available genomic and proteomic data from currently available databases must be screened.

Despite not being necessarily clade-specific, our set of genes is retained in very distantly related bilaterian species, while at the same time not being restricted to the group of classical, universally conserved housekeeping genes. This is interesting to note since even highly conserved and important genes can become lost, such as some Hox genes, e.g., from *C. elegans* [55] and the nematodes in general. Our set is also significantly divergent from potential non-bilaterian orthologs, suggesting either their origin or their rapid evolution in the ur-bilaterian. It thus appeared logically to ask which functions these orthologues might have in bilaterian animals. In comparison to classical enrichment analysis implemented in various software packages like blast2GO [56], our MSGEA test allows for the first time to compare GO-enrichment across several species. This is advantageous in the age of genomics where very often more than one species or several samples is analyzed. In our framework of three model bilaterians, a fruit fly, a zebrafish, and a roundworm, the MSGEA-based GO term analysis and subsequent literature mining indicated roles in development for the clustered genes. This link appears likely, since the process of development, the construction of adult specimens from single cells, is conserved in all animals and is also the phase when the combined traits of bilaterian life were most likely brought together: bilateral symmetry, triploblasty, the enhanced nervous system, and a set of complex cell types. In particular, the ‘muscle’ and ‘neuron’ class of clusters is compatible with the idea that key bilaterian innovations involved the central nervous system and muscles [57]. In general, the connection to development agrees with other studies, such as the large scale modEncode study of model organisms [46].

The conserved genes in set $L^{\prime }$ follow characteristic expression patterns during development, which are shared between species separated by a billion years of independent evolution. In particular, we observe characteristic low expression in early development and a peak towards the end of larval development common to highly derived taxa such as nematodes, insects and teleosts. The single (fish and nematode) and double peak (fly) patterns which we observed agree with results from the modEncode project [46,58]. Starting from co-expression analyses, the authors arrived at a set of orthologous genes acting in comparable stages of *D. melanogaster* and *C. elegans* development. The simpler, single peaked pattern in *C. elegans* and *D. rerio* reflects up-regulation at ventral enclosure and pharyngula stages, respectively. Thus, the expression profiles in these organisms mirror their life-cycle, displaying shared peaks when the adult body-plan is assembled. This phase, dominated by morphogenesis, has been interpreted as the phylotypic stage in both organisms [59,60]. In the holometabolous fly the situation is more complex: a body morphology is built twice, for the larva and then for the adult. These are two phases of increased cell proliferation during gastrulation and the transition from pupa to adult. The two peaks are also visible in the matched-function set, but they are less pronounced in the random-background set. It thus appears the two peaks we observed for our proteins reflect up-regulation during both periods of cell proliferation. It will be interesting to analyze if similar patterns can be observed in other species undergoing metamorphosis, particularly in vertebrates.

It has been hypothesized that differences in gene regulatory processes and the deployment of GRNs may have been more important for the evolution of Bilateria than the conservation of single key genes [61]. A conserved profile of developmental gene expression was observed in several studies across animal phyla, including in non-bilaterians, when developmental time course data were analyzed [12]. These studies suggest the existence of a phylotypic stage late in development, but see [12,48,62]. We observed low expression of young genes that originated after the emergence of Bilateria and of set $L^{\prime }$ genes in early development. We interpret this to indicate that early embryonic processes, e.g., initial cell divisions, are governed by evolutionary old genes, which are common to all bilateria and non-bilateria. Thus, while non-Bilateria have a similar complex distribution and abundance of gene regulatory elements and systems [63,64], and likely their own phylotypic stage, our data support the idea that a change in gene regulation in the phase when adult morphology is shaped might be linked to the emergence of Bilateria. Furthermore, the process appears to be conserved across immense evolutionary distances. These observations appear to also be in line with recently analyzed transcriptomic data suggesting a single and early invention of the metazoan larva, and a co-option of existing adult genetic programs into the larva stage [65].

## 5. Conclusions

In this study, we performed a systematic approach to identify bilaterian-specific genes in the face of constantly growing databases. Previous investigations concluded that essentially all important developmental regulators precede the origin of bilaterians, and that rewiring of already existing factors was a hallmark of bilaterian evolution [22,23,24,63,66] and our results enhanced by a second BLAST and phylogeny-based control step are in support of this. Future data from non-bilaterian species might in fact show that even the 35 groups of orthologues we now identified to be conserved across the three model species but without credible non-bilaterian orthologues were present before the split from Cnidaria. Conversely, it remains possible that genes independently lost in bilaterian crown groups have no non-bilaterian orthologues. Analyzing the roles of genes, we found conserved across crown groups in bilateria we found a strong link to development. Thus, our results indicate that Bilateria are unified by processes occurring late in embryonic development, which are shaped by a set of conserved genes. At the same time, early developmental processes might be shared across Metazoa. It thus seems important for evolutionary biologists to focus on processes—in particular in laboratory studies—to better understand the evolution of major taxonomic groups.

## Figures and Tables

**Figure 1 life-10-00182-f001:**
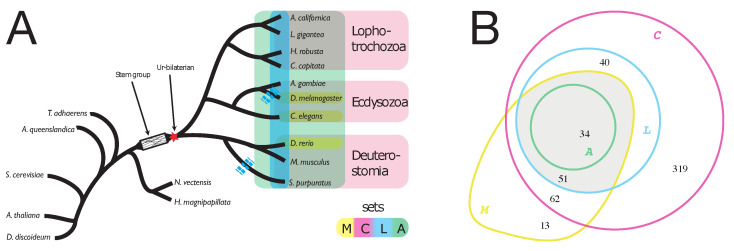
(**A**): Sketch of the genealogical relationship between the clades of Lophotrochozoa, Ecdysozoa and Deuterostomia and the species included in our analysis (Table 1). Bilateria emerged from unknown stem-group lineages in the early Cambrian. Legend: *M*: all model species are represented in a cluster; *C*: all clades are represented; *L*: at most one loss event along the tree (see Appendix A); *A*: all of nine bilaterian species are represented. Sketch of the “ur-bilaterion” after [4]. Blue dashed lines indicate examples of gene loss events. (**B**): Absolute numbers of clusters of orthologous proteins (COPs) in the four sets. Gray region: detailed analyses are performed for 85 COPs in set $L^{\prime }=L\cap M$.

**Figure 2 life-10-00182-f002:**
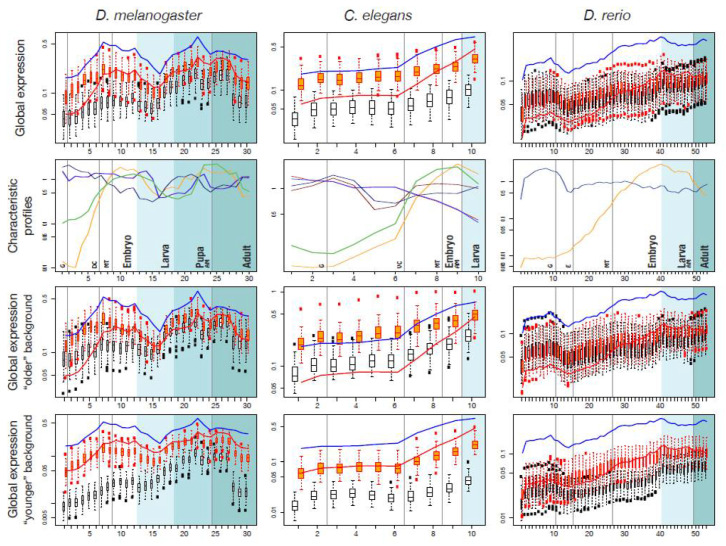
Conservation scores for orthologous exons from PhastCons analysis. For each species, we show the correlation between scores (**top**), the distribution of the fraction of very conserved sites (**middle**), and of the average conservation score (**bottom**) among exons.

**Figure 3 life-10-00182-f003:**
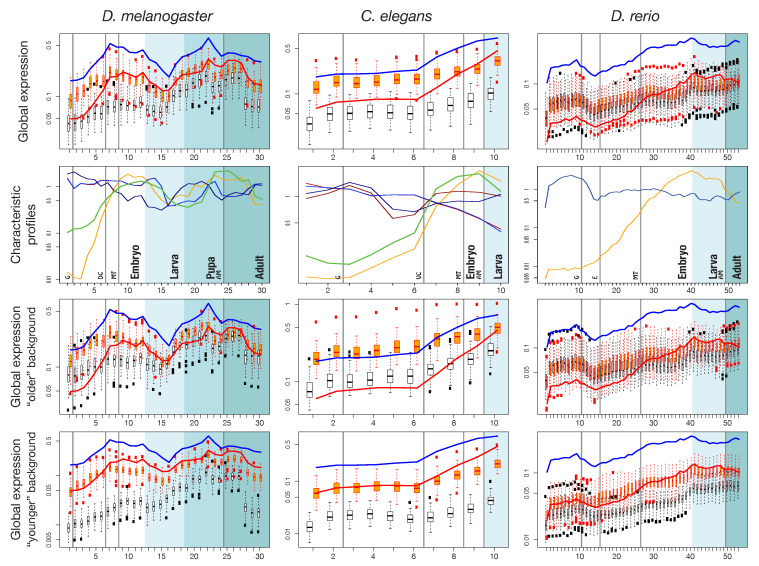
Expression profiles of different species. Top row: average (global) expression profiles for genes collected in set $L^{\prime }$. Red line: pooled profiles for “all orthologs and paralogs” (set ”AO”). Blue line: profiles for the “most conserved orthologs” (set “MCO”). The white and orange boxplots show the “random” and “matched-function” expression backgrounds, respectively (see text). The most appropriate comparison is between AO and matched-function background (red and orange respectively). Second row: Characteristic expression profiles for Bilateria, i.e., expression profiles that are over-represented in set L’ compared to the rest of the genome. Different colors represent different clusters. Darker and lighter colors represent profiles which are similar among species. Bottom rows: distributions shown in boxplots are calculated from genes which are younger (phylostrata ≥7 according to the terminology of [43]) or older (phylostrata <7) than bilateria. Background shading (from white to dark grey) represents developmental stages embryo, larva, pupa (fly only) and adult. For a detailed explanation of the developmental stages see Appendix A. Meaning of labels in row 2: G-gastrulation; DC-dorsal closure; MT-first muscle twitching; AM-transition to adult morphology; VC-ventral closure; E-100% epiboly

**Figure 4 life-10-00182-f004:**
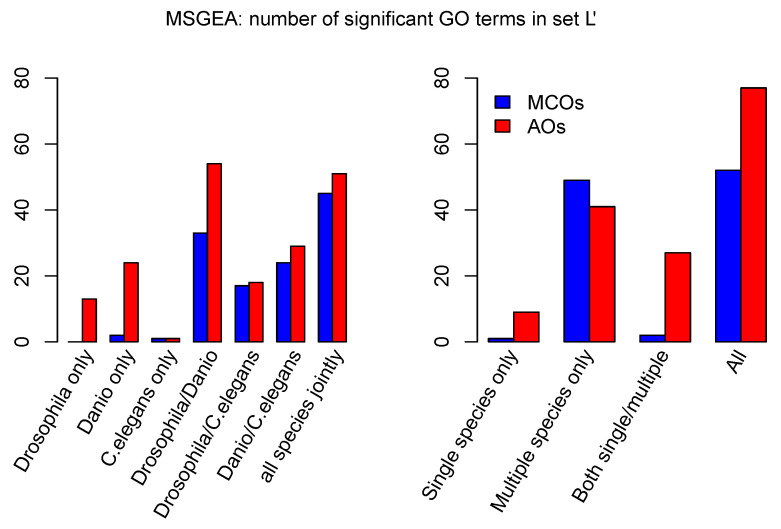
Number of significant GO terms identified with MSGEA in set $L^{\prime }$ (p<0.05). Red histograms: considering all orthologs (AO) from a cluster; blue histograms: only the most conserved orthologs (MCO) are considered in the MSGEA analysis. Left: number of significant terms found for each single- or multi-species enrichment analysis. Right: number of significant terms found only in single/multiple-species enrichment analyses and in both kind of analyses. The large amount of GO terms found only in multiple-species analysis illustrates the power of MSGEA.

**Figure 5 life-10-00182-f005:**
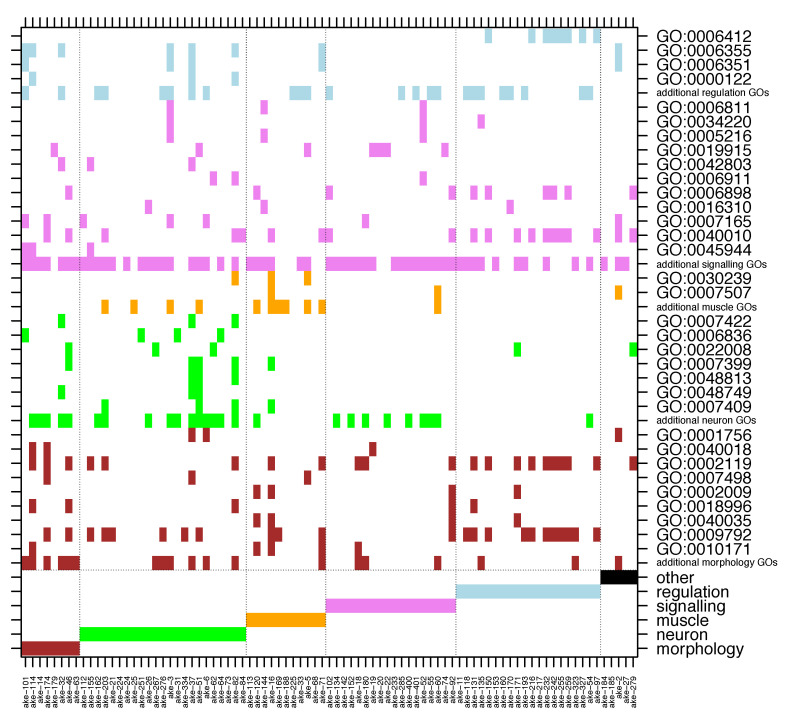
Significant GO terms identified in set $L^{\prime }$ by applying MSGEA (see text). The upper part of the figure shows the GO terms which occur in each COP (*x*-axis). Those terms which occur at least in three COPs are spelled out with their GO-ID, otherwise they are collected in the lines ‘additional … GOs’. Only non-generic GO terms of level 4 and higher are recorded. The bottom six rows show the assignment of COPs to functional classes based on human-curated literature mining. As can be seen from the color code, automatic GO and human-curated functional assignments do not always coincide.

**Table 1 life-10-00182-t001:** Species considered to contrast bilaterian and non-bilaterian protein repertoires

Species	Data Source	No. Proteins in Database	No. Proteins in all COPs	No. COPs w/ Species ${}^{1}$	No. Proteins per COP ${}^{2}$
**Bilateria**					
Deuterostomia					
*Danio rerio*	Ensembl	41,693	1190	428	2.78
*Strongylocentrotus purpuratus*	NCBI	42,420	652	260	2.51
*Mus musculus*	Ensembl	40,732	867	357	2.43
Ecdysozoa					
*Anopheles gambiae*	Ensembl	13,133	380	307	1.24
*Drosophila melanogaster*	FlyBase	23,849	989	392	2.52
*Caenorhabditis elegans*	WormBase	25,634	602	291	2.07
Lophotrochozoa					
*Aplysia californica*	NCBI	1093	15	7	2.14
*Capitella teleta*	JGI	32,415	642	376	1.71
*Helobdella robusta*	JGI	23,432	420	325	1.29
*Lottia gigantea*	JGI	23,851	492	337	1.46
**Non-Bilateria**					
*Amphimedon queenslandica*	NCBI	9908			
*Arabidopsis thaliana*	TAIR	28,952			
*Dictyostelium discoideum*	Dictybase	13,426			
*Hydra magnipapillata*	NCBI	17,563			
*Nematostella vectensis*	JGI	27,273			
*Saccharomyces cerevisiae*	SGD	6692			
*Trichoplax adhaerens*	JGI	11,520			

${}^{1}$: number of COPs (out of |C∪M|=519), in which a given species is represented; ${}^{2}$: average number of paralogs within a COP (number of proteins/number of COPs).

**Table 2 life-10-00182-t002:** COP distribution in different data sets.

Species	Set *C* (506)			Set *M* (160)			Set *L* (125)			L∩M (85)			Set *A* (34)		
	#P	#C	Ratio	#P	#C	Ratio	#P	#C	Ratio	#P	#C	Ratio	#P	#C	Ratio
Deuterostomia															
*D. rerio*	1169	415	82.0	690	160	100	555	125	100	436	85	100	205	34	100
*S. purpuratus*	636	254	50.2	233	89	55.6	288	103	82.4	175	63	74.1	90	34	100
*M. musculus*	855	349	69.0	398	134	83.8	428	121	96.8	301	81	95.3	163	34	100
Ecdysozoa															
*A. gambiae*	373	300	59.3	156	120	75.0	143	112	89.6	107	78	91.8	51	34	100
*D. melanogaster*	964	379	74.9	494	160	100	387	116	92.8	317	85	100	143	34	100
*C. elegans*	589	278	54.9	359	160	100	231	94	75.2	207	85	100	101	34	100
Lophotrochozoa															
*C. teleta*	642	376	74.3	247	116	72.5	198	119	95.2	153	79	92.9	65	34	100
*H. robusta*	420	325	64.2	165	109	68.1	160	115	92.0	110	75	88.2	48	34	100
*L. gigantea*	492	337	66.6	224	117	73.1	217	121	96.8	169	81	95.3	84	34	100
*[A. californica]*	15	7	01.4	7	4	2.5	3	2	1.6	3	2	2.4	1	1	2.9

#P: number of proteins from given species in given set. #C: number of COPs (out of a maximum number *x* given in parentheses in the header line) which contain at least one protein from a given species. Ratio: percentage #C/x·100.

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
