# Peer review of "Conserved Patterns in Developmental Processes and Phases, Rather than Genes, Unite the Highly Divergent Bilateria"

_life, 2020, doi:10.3390/life10090182_

Round 1

Reviewer 1 Report

See attached

Reviewer 2 Report

This paper describes a bioinformatics-based analysis of bilaterian and non-bilaterian genes, which initially aimed at identifying bilaterian-specific genes. The authors obtained a set of 85 gene orthogroups that are present in all of D. melanogaster, C. elegans, and D. rerio but not in several non-bilaterian species. However, their arguments are not straightforward because they understand that the "bilaterian-specific" gene list can be affected by thresholds, genome annotations, and information available from non-bilaterian species. Then they introduced a novel multi-species GO-enrichment method to obtain enriched GO terms for the "bilaterian-specific" gene set, which suggest roles in development. They also analyzed developmental expression profiles of "bilaterian-specific" genes to detect some similarities across the three model species. Based on detected similar peaks, the authors argue for the presence of the phylotypic stage in the divergent bilaterian species. Finally, they showed that the expression profiles of "bilaterian-specific" genes are more similar to those of younger genes than to those of older genes. Although results of the analyses may have some significant signs, I do not feel that the findings and logical flows to reach conclusions reflected by the paper title are sufficiently strong. I have several major concerns about analysis designs and data interpretations, as follows:

1) Currently there is a lot of genome information available from many bilaterian species. Why did the authors use a handful set of such data? The selection of the species is highly biased. How do these biases affect the gene content of set L'? It is generally recognized that the Ecdysozoan models experienced gene content reductions. If more species with ancestral genomes are included in the analyses, what happens?

2) In the latter half of this study, analyses were focused on the gene set L', in which pseudo bilaterian-specific genes might be included. It is difficult to evaluate the significance of the data derived from the gene set L' in the line of bilaterian evolution.

3) GO enrichment analyses were applied for the gene set L'. However, to detect specific evolutionary signs for the bilaterians, a "non-bilaterian metazoan specific" gene set and/or other control gene sets should be analyzed and compared.

4) Fig5 top row: Expression profiles for genes in set L' are compared with two kinds of background profiles. Honestly, I cannot identify differences of set L' from the backgrounds and similarities across the species. What aspects can we extract from the graphs?

5) Fig5 2nd row: These data may be significant, at least when compared between D. melanogaster and C. elegans. However, no clues (only colors) are provided to know what gene clusters represent the characteristic profiles. If more information (e.g. gene function and orthologs from other species) is provided for these gene clusters, the analyses may achieve a broader significance. However, it is still difficult to find similarities between D. melanogaster/C. elegans and D. rerio.

6) I do not feel that the paper title is appropriate. What does "Developmental genetics" mean? No genetics was used in this work. The title should be more directly related to the results presented.

7) The title and abstract stress the finding of conservation between highly divergent bilaterian species. However, none of the species names is described there. This may cause expectations.

Round 2

Reviewer 2 Report

I find that the manuscript has been improved following the suggestions.